# Diagnostic Performance of Two-Dimensional Ultrasound, Two-Dimensional Sonohysterography and Three-Dimensional Ultrasound in the Diagnosis of Septate Uterus—A Systematic Review and Meta-Analysis

**DOI:** 10.3390/diagnostics13040807

**Published:** 2023-02-20

**Authors:** Juan Luis Alcázar, Isabel Carriles, María Belén Cajas, Susana Costa, Sofia Fabra, Maria Cabrero, Elena Castro, Aida Tomaizeh, María Victoria Laza, Alba Monroy, Irene Martinez, Maria Isabel Aguilar, Elena Hernani, Cristina Castellet, Agustin Oliva, María Ángela Pascual, Stefano Guerriero

**Affiliations:** 1Department of Obstetrics and Gynecology, Clinica Universidad de Navarra, 31008 Pamplona, Spain; 2Department of Obstetrics and Gynecology, Hospital Universitario de Salamanca, 37007 Salamanca, Spain; 3Department Obstetrics and Gynecology, Centro Hospitalar e Universitário de São João, 4200-319 Porto, Portugal; 4Department of Obstetrics and Gynecology, Hospital Universitario Infanta Sofia, 28702 Madrid, Spain; 5Department of Obstetrics and Gynecology, Hospital Universitario Virgen de la Arrixaca, 30120 Murcia, Spain; 6Department of Obstetrics and Gynecology, Hospital Universitario Virgen de Valme, 41701 Sevilla, Spain; 7Department of Obstetrics and Gynecology, Hospital Universitario Materno-Infantil, 06010 Badajoz, Spain; 8Department of Obstetrics and Gynecology, Hospital General Universitario de Castellón, 12004 Castellón, Spain; 9Department of Obstetrics, Gynecology, and Reproduction, Hospital Universitari Dexeus, 08028 Barcelona, Spain; 10Department of Obstetrics and Gynecology, Hospital Universitario San Carlos, 28040 Madrid, Spain; 11Centro Integrato di Procreazione Medicalmente Assistita (PMA) e Diagnostica Ostetrico-Ginecologica, Azienda Ospedaliero Universitaria-Policlinico Duilio Casula, Monserrato, 09042 Cagliari, Italy; 12Dipartimento di Scienze Chirurgiche, University of Cagliari, 09124 Cagliari, Italy

**Keywords:** ultrasound, sonohysterography, septate uterus, mullerian anomalies

## Abstract

Background: The septate uterus is the most common congenital uterine anomaly, and hysteroscopy is the gold standard for diagnosing it. The goal of this meta-analysis is to perform a pooled analysis of the diagnostic performance of two-dimensional transvaginal ultrasonography, two-dimensional transvaginal sonohysterography, three-dimensional transvaginal ultrasound, and three-dimensional transvaginal sonohysterography for the diagnosis of the septate uterus. Methods: Studies published between 1990 and 2022 were searched in PubMed, Scopus, and Web of Science. From 897 citations, we selected eighteen studies to include in this meta-analysis. Results: The mean prevalence of uterine septum in this meta-analysis was 27.8%. Pooled sensitivity and specificity were 83% and 99% for two-dimensional transvaginal ultrasonography (ten studies), 94% and 100% for two-dimensional transvaginal sonohysterography (eight studies), and 98% and 100% for three-dimensional transvaginal ultrasound (seven articles), respectively. The diagnostic accuracy of three-dimensional transvaginal sonohysterography was only described in two studies, and we did not calculate the pooled sensitivity and specificity for this method. Conclusion: Three-dimensional transvaginal ultrasound has the best performance capacity for the diagnosis of the septate uterus.

## 1. Introduction

Congenital uterine anomalies (CUAs) of the genital tract are the result of abnormal formation, canalization, or fusion of the paramesonephric ducts or the defective absorption of the midline septum during fetal life [1]. Prevalence in low-risk populations is difficult to assess, mostly because diagnostic methods are rarely applied in asymptomatic populations and there is a lack of universally applicable criteria [2]. However, the systematic review published by Chan et al. found a prevalence of 5.5% in unselected populations, 8% in infertile women, 13.3% in those with miscarriage, and 24.5% in infertile women who also had a history of miscarriage [3].

The septate uterus is the most common uterine anomaly, accounting for 35% of all CUAs, with a prevalence of 0.2–2.3% in women of reproductive age [4]. Women with a septate uterus have been reported to have a higher risk of infertility, recurrent miscarriage, preterm birth, and fetal malpresentation [5]. The etiopathogeny of the worse reproductive outcomes in women with septate uteri has not yet been fully explained. The presence of a septum itself and some histological changes in the endometrium, such as the lower expression of HOXA10 genes and VEGF receptor genes and a lower existence of glandular and ciliated cells, have been proposed as possible explanations [6].

The criteria used to diagnose uterine anomalies in general and the septate uterus in particular are a subject of debate. There are two main classification systems: the American Society of Reproductive Medicine (ASRM) classification, published by the American Fertility Society in 1988 and modified in 2016 and 2021 [7], and the ESHRE-ESGE one, published in 2013 by the European Societies of Endoscopic and Reproductive Gynecology [8]. Neither of them is universally accepted. According to the ASRM 2021 guideline, a septate uterus has an indentation depth > 1 cm and an indentation angle < 90° with a normal external shape. ESHRE-ESGE considers an indentation-to-wall-thickness (I:WT) > 50%. This lack of unanimously accepted criteria confuses both patients and healthcare professionals in terms of diagnosis and potential treatments.

On the other hand, hysteroscopic septum resection is a widespread technique throughout the world with the aim of improving reproductive outcomes. However, the evidence supporting this procedure is controversial, with many studies lacking enough scientific quality [9,10]. The different medical societies around the globe provide different recommendations. The ASRM guideline recommends septum resection, but the ESHRE guidelines do not, highlighting the need for higher-quality studies. Recently, high-quality articles with contradictory conclusions have been published. While the international multicenter randomized controlled trial published by Rikken et al. showed no improvement in live births or any other reproductive outcomes related to hysteroscopic septum resection, the meta-analysis published by Carrera et al. supports that this surgical technique is effective in reducing the risk of miscarriage in patients with a complete or partial uterine septum [11,12]. An accurate diagnosis is needed to offer these women the right management.

In addition to the drawbacks of the previous classification systems, the wide variety of innovative, non-invasive, high-accuracy diagnostic techniques available also make the categorization of CUAs a clinical challenge. Two-dimensional transvaginal ultrasonography (2D TVS) is the most available diagnostic technique, and it is routinely used as the first approach in the diagnosis of CUAs; nevertheless, differentiating between the different types of anomalies is challenging with this method [13]. Magnetic resonance imaging (MRI) has also become a reliable diagnostic approach; however, it is more expensive and less available than an ultrasound [14,15]. Three-dimensional transvaginal ultrasound (3D TVS) can provide highly objective and, most importantly, measurable information for the anatomy of the cervix, uterine cavity, uterine wall, and external contour of the uterus. Displaying the coronal plane can provide a clear image of the cavity and external profile of the uterine fundus.

Although hysteroscopy is currently considered the gold standard in the diagnosis of CUAs [16], 3D TVS has been suggested as the preferred non-invasive diagnosis technique due to its high accuracy and accessibility [2,3,8]. To the best of our knowledge, no previous studies about the diagnostic performance of 2D TVS, 2D transvaginal sonohysterography (2D SIS), 3D TVS, and 3D transvaginal sonohysterography (3D SIS) in the evaluation of women with a septate uterus have been published. The main objective of this article is to perform a pooled analysis of the techniques mentioned above for the specific diagnosis of a septate uterus.

## 2. Materials and Methods

### 2.1. Protocol and Registration

This meta-analysis was conducted following the PRISMA (Preferred Reporting Items for Systematic Reviews and Meta-analysis) recommendations and the SEDATE (Synthesizing Evidence from Diagnostic Accuracy Tests) guidelines [17,18]. Inclusion and exclusion criteria were defined prior to starting data research. The study protocol was not registered in PROSPERO. There was no need for an ethics committee’s approval given the nature and design of this study.

For this meta-analysis, we considered as septate uteri those cases reported as septate (noted as complete or partial “subseptate” in many studies) in the primary studies included.

### 2.2. Data Sources and Searches

Five authors screened three electronic databases (PubMed/Medline, Scopus, and Web of Science). The period was set between January 1990 and November 2022. The language was set to English only. The terms used for both searches were as follows: “uterine anomalies,” “müllerian anomalies,” “transvaginal ultrasound,” and “sonohysterography.”

### 2.3. Study Selection and Data Collection

In collaboration, the five authors combined the searches in different databases and excluded duplicated articles. In the next step, we filtered the titles first and the abstracts second to identify irrelevant articles to exclude, such as those not strictly related to the topic under review or non-observational studies (i.e., reviews, case reports, and letters to the editor). Records were then filtered again with a complete reading of the full text of the studies that remained after exclusions.

This meta-analysis had the following inclusion criteria:-Prospective or retrospective observational cohort studies including women diagnosed with uterine Müllerian anomalies using any of the following methods as index tests: 2D transvaginal ultrasound, 2D sonohysterography, 3D transvaginal ultrasound, or 3D sonohysterography.-Data reported that allows for the construction of a 2 × 2 table to estimate true positive, true negative, false positive, and false negative cases for any of the index tests assessed.-Hysteroscopy, with or without combined laparoscopy, as the reference standard-The exclusion criteria were:-Studies not related to the topic-Articles not reporting specific data regarding the septate uterus (complete or partial)-Letters to the editor, commentaries, narrative reviews, consensus documents, and any other study that does not provide enough data to construct a 2 × 2 table-Hysteroscopy with or without combined laparoscopy is not used as the gold standard for the diagnosis of uterine Müllerian anomalies.

If the dates of two cohort studies published by the same authors overlapped, we excluded the first one in order to avoid the inclusion of duplicate cohorts. We did not contact the authors. We used the *snowball* strategy to identify potential interesting papers by reading the reference lists of the papers selected for full text reading.

Five authors independently retrieved the following data from each study: first author, year of publication, country, study design, number of centers participating, patients’ inclusion criteria, patients’ exclusion criteria, patients’ age, number of patients, number of patients with septate uterus, index test used, definition of septate used, number of examiners, whether the examiner was blinded or not to the reference standard, the reference standard used, the diagnostic accuracy results, and the time elapsed from the index test to the reference standard test. Disagreements arising during the process of study selection and data extraction were resolved by consensus among these five authors.

### 2.4. Risk of Bias in Individual Studies

The quality assessment of the studies included in the meta-analysis was conducted using the tool provided by the Quality Assessment of Diagnostic Accuracy Studies-2 (QUADAS-2) [19]. The QUADAS-2 format includes four domains: patient selection, index test, reference standard, and flow and timing. For each domain, the risk of bias and concerns about applicability (not applying to the domain of flow and timing) were analyzed and rated as low, high, or unclear risk. Five authors independently evaluated the methodological quality. Disagreements were solved by discussion between these authors.

The assessment of the quality was based on whether the article described the study’s design, inclusion and exclusion criteria, if the operators were blinded, whether the study reported on how the index test was performed and interpreted, which was the reference standard used, and a description of the time elapsed from the index test assessment to the reference standard result. Unclear risk was stated when the corresponding information for each domain was not reported in the study. The arcuate uterus was considered a variant of normality and, therefore, was not considered a Müllerian anomaly. For the papers that studied the arcuate uterus separately, authors included them in the non-septate group.

Surgery, including hysteroscopy and/or laparoscopy, was defined as the reference standard. The fact that surgeons were not blinded to ultrasound findings was not considered to pose a high risk of bias. For the flow-and-timing domain, we considered a high risk of bias as time elapsed from the index test assessment to the reference standard. We decided to consider all studies that reported this data as low-risk since the uterine septum is present, by definition, since patient birth and it is assumed that it does not change over time.

### 2.5. Statistical Analysis

Heterogeneity for sensitivity and specificity was assessed using Cochran’s Q statistic and the I^2^ index. A *p*-value < 0.1 indicates heterogeneity. I^2^ values of 25%, 50%, and 75% would be considered to indicate low, moderate, and high heterogeneity, respectively [20]. Forest plots of the sensitivity and specificity of all studies were plotted. Meta-regression was used if heterogeneity existed to assess covariates that could explain this heterogeneity. The co-variates analyzed for meta-regression were year of publication, sample size (n), uterine septum prevalence, classification type used, and type of study design.

Summary receiver operating characteristic (sROC) curves were plotted to illustrate the relationship between sensitivity and specificity, and the area under the curve (AUC) was calculated. Publication bias was assessed using Deek’s method [21].

Statistical analysis was performed using STATA version 12.0 for Windows (Stata Corporation, College Station, TX, USA). A *p*-value < 0.05 was considered statistically significant.

## 3. Results

### 3.1. Search Results

The electronic search provided 897 citations. We excluded 176 duplicate records, and 721 citations remained. After reading titles and abstracts, 675 citations were excluded (papers not related to the topic (n = 536), reviews (n = 78), case reports, letters to the editor, and commentaries (n = 39), and papers written in a non-English language (n = 22)). Forty-six papers remained for full-text reading, and, after that, 32 papers were excluded (no data about the septum (n = 21), no data about malformations (n = 1), hysteroscopy was not the gold standard (n = 4), no data available to construct a 2 × 2 table (n = 2), reviews, or letters to the editor (n = 4)). Four additional studies were identified using the snowball strategy. Eighteen studies were ultimately included in this meta-analysis for qualitative and quantitative synthesis [13,22,23,24,25,26,27,28,29,30,31,32,33,34,35,36,37,38]. A flowchart summarizing the literature search is shown in Figure 1.

### 3.2. Characteristics of Included Studies

The PICOS features of the included studies are given in Table 1.

Out of the eighteen studies included in this meta-analysis, ten studies assessed 2D TVS [13,22,23,24,26,28,32,34,36,37], eight studies assessed 2D SIS [13,23,24,27,28,29,31,36], seven studies assessed 3D TVS [13,24,25,30,34,35,38], and two studies assessed 3D SIS [13,33] (Table 1). These papers were published between March 1993 and March 2019 and reported the data of 3737 women with ages ranging from 18 to 49 years old in the final analysis. In 638 of these, the uterine septum was identified by hysteroscopy. The mean prevalence of uterine septum was 27.8% (range 2.9–88.0%).

### 3.3. Methodological Quality of Included Studies

Concerning the studies included in this meta-analysis: seven of them were prospective [13,27,30,31,36,37,38], four were retrospective [23,28,34,35], two were cross-sectional [32,33], and in the remaining five studies this issue was not specified [22,24,25,26,29]. The examiners performing hysteroscopy were blinded to the results of the index tests in nine studies [13,23,27,30,31,32,36,37,38] and in the other nine this subject was not specified [22,24,25,26,28,29,33,34,35].

The definition of uterine septum was not specified in four studies [25,26,29,36], six studies described their own definition of uterine septum [22,23,27,28,31,37], seven articles classified uterine septum according to ASRM criteria [13,24,30,32,33,34,35], and one used the ESHRE-ESGE definition [38].

The risk of biased evaluation and concerns regarding the applicability of the selected studies can be seen in Table 2.

Thirteen studies were considered high-risk in the “patient selection” domain because either the inclusion criteria [13,23,25,28,31,32,34,35,36,37,38] (abnormal uterine bleeding, cervical polyps, CUAs suspected) or the exclusion criteria [13,25,30,32,34,35,36,37,38] (no accurate visualization of the endometrium, patients who do not want to become pregnant, uterine fibroids or endometrial polyps detected at ultrasound, women with previous uterine surgery with heterogenic or echogenic endometrium due to bleeding, chronic systemic illness, suspect of fusion anomalies such as bicorporeal or bicornuate uterus, and women 40 years old or older) were insufficient.

Concerning the domain “index test,” three studies were considered to have a high risk of bias because the definition applied to describe the uterus septate was not adequate. The definitions “myometrial echoes divided the fundal endometrial image in the transverse plane” [22] and “abnormal uterine cavity shape” [23] do not distinguish between uterus septate or uterus didelphys/bicorporeal, and “abnormal uterine contour” [27] does not occur only in the uterine septum. Four other studies were of unclear risk of bias for the “index test” because the authors did not explain what uterine septum diagnostic criteria were used [25,26,29,36].

All the studies were considered low-risk for the domain “reference standard,” since all of them use hysteroscopy with or without laparoscopy as a reference standard to detect the uterine septum as determined in our inclusion criteria.

Regarding the domain “flow and timing,” six articles were considered to have an unclear risk of bias because the time elapsed between the ultrasound and HSC was not described [13,28,29,31,35,38]. In the remaining twelve studies, it was considered low risk [22,23,24,25,26,27,30,32,33,34,36,37].

For the analysis of concerns about applicability, one study [23] was considered to have a high risk of bias for “patient selection” because women with abnormal uterine bleeding are not an adequate target population to investigate the uterine septum. All the other studies’ domains were considered low-risk.

### 3.4. Diagnostic Performance of 2D TVS and 2D SIS for Uterine Septum Detection

Concerning 2D TVS performance for diagnosing a septate uterus, the pooled sensitivity was 83% (95% CI 64–93%) with a high degree of heterogeneity (I^2^ = 94.7%; 95% CI 92.6–96.8%; Cochran Q = 169.4, *p* < 0.001). The estimated pooled specificity was 99% (95% CI 94–100%) with high heterogeneity (I^2^ = 93.3%; 95% CI 90.5–96.2%; Cochran Q = 135.1, *p* < 0.001). The forest plot is shown in Figure 2.

The estimated pooled sensitivity for 2D SIS was 94% (95% CI 66–99%) with a high degree of heterogeneity (I^2^ = 90.78%, 95% CI 85.84–95.72%; Cochran Q = 75.94, *p* < 0.001). The estimated pooled specificity was 100% (95% CI 99–100%) with moderate heterogeneity (I^2^ = 55.52%, 95% CI 20.30–90.75%; Cochran Q = 15.74, *p* < 0.001). The forest plot is shown in Figure 3.

Meta-regression showed that for 2D TVS, uterine septum prevalence explained the heterogeneity observed in uterine septum diagnosis (*p* < 0.05), but the publication year and sample size did not. Regarding 2D SIS, the year of publication, sample size, and uterine septum prevalence did not explain the heterogeneity observed in diagnostic performance.

The ROC curves for the diagnostic performance of the 2D TVS and 2D SIS to detect uterine septum are shown in Figure 4 and Figure 5, respectively. The area under the curve regarding 2D TVS was 0.98 (95% CI 0.96–0.99). For 2D SIS, the area under the curve was 1.00 (95% CI 0.99–1.00). Fagan’s monogram showed that 2D TVS increased the pre-test probability of a uterine septum from 23% to 97% and decreased to 5%, with an LR+ and LR− of 107 and 0.18, respectively (Figure 6). On the other hand, 2D SIS increased the pre-test probability of a uterine septum from 27% to 99% and decreased to 2%, with an LR+ and LR− of 455 and 0.06, respectively (Figure 7). We did not find publication bias referred to 2D TVS (*p* = 0.71) and 2D SIS (*p* = 0.06) (Figure 8 and Figure 9, respectively).

### 3.5. Diagnostic Performance of 3D TVS and 3D SIS for Uterine Septum Detection

There are only two articles [13,33] describing the diagnostic accuracy of 3D SIS, and, for this reason, we cannot calculate the pooled sensitivity and specificity of this method.

Regarding 3D TVS to diagnose the uterine septum, pooled sensitivity was 98% (95% CI 95–99%) with a high degree of heterogeneity (I^2^ = 94.0%; 95% CI 90.9–97.1%; Cochran Q = 100.0, *p* < 0.001). Pooled estimated specificity was 100% (95% CI 78–100%) with high heterogeneity (I^2^ = 97.6%; 95% CI 96.6–98.5%; Cochran Q = 247.3, *p* < 0.001). The forest plot is shown in Figure 10. Meta-regression showed that the year of publication, sample size, and uterine septum prevalence did not explain the heterogeneity observed in diagnostic performance.

The ROC curve for the diagnostic performance of the 3D TVS to detect uterine septum is shown in Figure 11. The area under the curve was 0.99 (95% CI 0.94–1.00). Fagan’s monogram showed that 3D TVS increased the pre-test probability of a uterine septum from 67% to 100% and decreased to 5% (Figure 12), with an LR+ and LR− of 504 and 0.02, respectively. We did not find publication bias regarding 3D TVS (*p* = 0.41), as shown in Figure 13.

## 4. Discussion

### 4.1. Summary of Evidence

In the present study, we performed a meta-analysis of the diagnostic performance of the 2D TVS, 2D TV SIS, and 3D TVS in the detection of the septate uterus, compared to the actual gold standard: the hysteroscopy and/or laparoscopy. We found 18 studies composed of 3737 patients with available data for analysis. The prevalence of the septate uterus was 27.8% (1039 cases) in a population of mostly infertile patients.

In our meta-analysis, the pooled sensitivity and specificity were 83% and 99% for 2D TVS, 94% and 100% for 2D SIS, and 98% and 100% for 3D TVS, respectively. The majority of the studies were prospective. Additionally, seven studies used the American Society for Reproductive Medicine (ASRM) criteria [13,24,30,32,33,34,35], one used the ESHRE-ESGE criteria [38], and the rest used their own definition or it was unclear.

### 4.2. Limitations and Strengths

The main strength of our study is that this meta-analysis is the first to address this issue. We believe that the methodology used is correct.

As limitations of our meta-analysis, we consider that the number of studies was low, as well as the quality of some papers; therefore, the results should be taken with caution. The characteristics of the participants included are part of this statement. The majority of the patients had a history of infertility, which we consider to be the most relevant group of women to study, although there is one article that used any patient for whom a hysteroscopy was indicated (due to infertility, but also abnormal uterine bleeding or cervical polyps). Furthermore, a sub-analysis of these patients was not performed.

Additionally, seven studies only recruited patients with a suspected mullerian anomaly based on a previous 2D TVS evaluation [13,25,28,31,34,35,38] and another two excluded patients older than 40 years [25,36], a decision that we consider a selection bias. Another limitation was the lack of uniformity of ultrasound definitions to classify the CUAs, which could lead to unnecessary inclusions or exclusions in the studies.

### 4.3. Interpretation of Results

According to our data, 2D SIS showed a better sensitivity than 2D TVS (94% vs. 83%) and almost the same specificity (99% vs. 100%). On the other hand, 3D TVS has the best performance capacity for the diagnosis of the septate uterus (98% sensitivity and 100% specificity).

2D SIS has demonstrated its utility for the evaluation of the uterine cavity; it is accessible to most centers and has mild and uncommon side effects [39]. Therefore, in a center without a 3D TVS, the 2D SIS is a feasible and accessible option for the diagnosis of the septate uterus. Both techniques are significantly less expensive than others, such as an MRI.

As stated above, the prevalence of the septate uterus in this meta-analysis was 27.8%. However, the studies included did not use the same definition criteria because, to date, there is not a universally accepted classification system for CUAs, which is problematic because, depending on the definition used, there is a different prevalence of findings. Over time, some scientific societies have made efforts to solve this issue. However, there is no consensus on this topic, and the different definitions and their impact on diagnosis have been proven in several studies [40,41,42,43].

Congenital Uterine Malformation by Experts (CUME) published an article in 2018 with the goal of assessing the level of agreement between experts in distinguishing a septate uterus from a normal or arcuate uterus. It was discovered that, over a series of patients, a uterus was defined as having a septum seven times more frequently using the ESHRE-ESGE criteria than the ASRM one. According to this statement, the ESHRE-ESGE criteria overdiagnose the septate uterus, while the ASRM criteria underdiagnose it [40]. CUME defines new cut-off values for indentation depth (≥10 mm), indentation angle (< 140°), and I:WT ratio (>110%) and suggests using internal indentation depth to distinguish between normal/arcuate and septate uteri because of its simplicity and reliability [40]. Therefore, the decision to treat a septate uterus that might also be classified as normal is challenging for the physician. This problem also has repercussions for evaluating the clinical impact of different CUAs; thus, comparisons between management are limited.

### 4.4. Future Research Agenda

There is a need for better quality studies that use uniform definition criteria in a larger series of patients to definitively present the performance of the different diagnosis methods, maybe including others such as the MRI. We consider that a cost-efficiency analysis could be performed in order to determine recommendations in this area.

## 5. Conclusions

Regarding these results and because the 2D TV SIS is an invasive method, we can conclude that 3D US is the best method for the diagnosis of the septate uterus, but as this equipment is not available in all centers, 2D SIS could be a reasonable option.

## Figures and Tables

**Figure 1 diagnostics-13-00807-f001:**
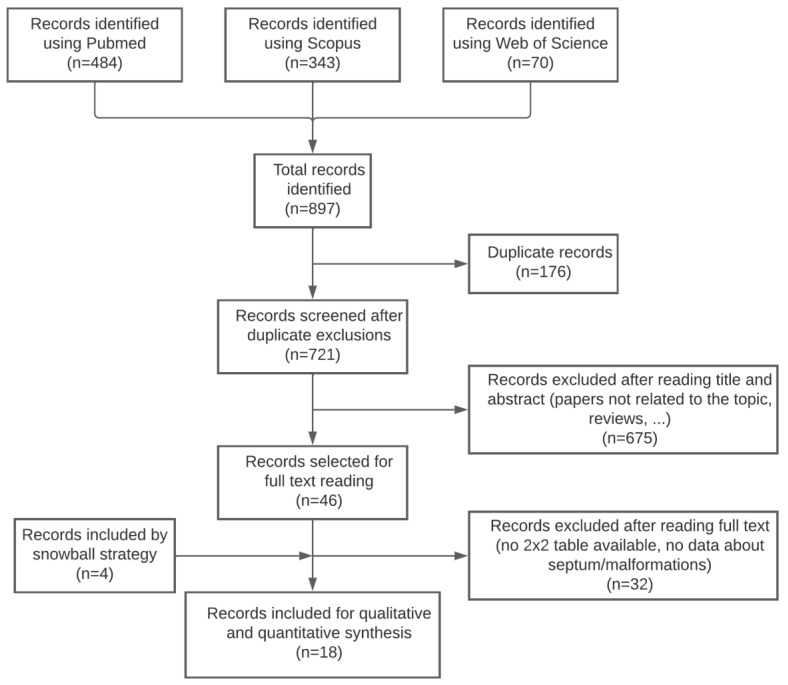
Flowchart showing the study selection process, indicating the titles found in each database, the exclusion process, and the final number of articles included in the meta-analysis.

**Figure 2 diagnostics-13-00807-f002:**
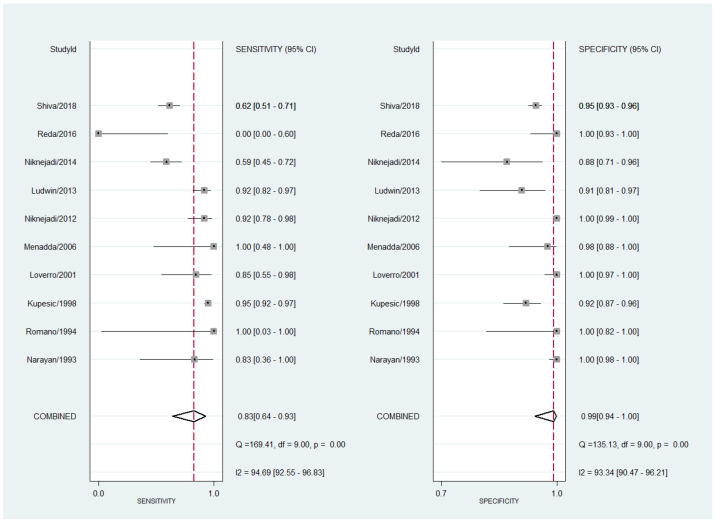
Forest plot for sensitivity and specificity for all studies concerning the diagnostic performance of two-dimensional transvaginal ultrasonography for uterine septum detection [13,22,23,24,26,28,32,34,36,37].

**Figure 3 diagnostics-13-00807-f003:**
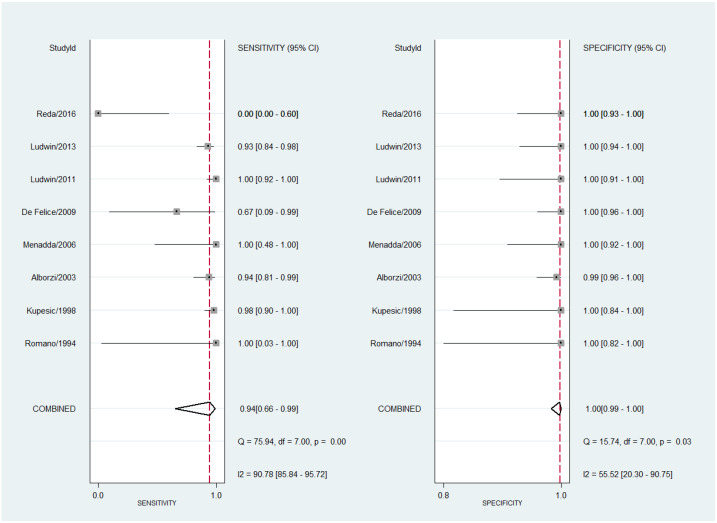
Forest plot for sensitivity and specificity for all studies concerning the diagnostic performance of two-dimensional sonohysterography for uterine septum detection [13,23,24,27,28,29,31,36].

**Figure 4 diagnostics-13-00807-f004:**
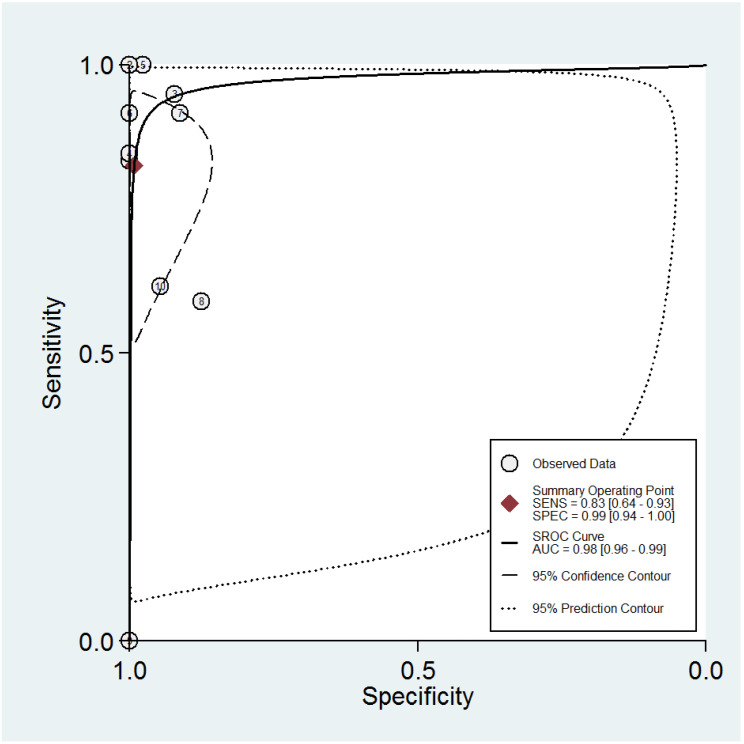
Summary ROC curve for the diagnostic performance of two-dimensional transvaginal ultrasonography to detect uterine septum, showing the sensitivity and specificity for each study and pooled estimation. The dashed line around the summary point estimate (red diamond) represents the 95% confidence region. The dotted line showing the 95% prediction contour corresponds to the predicted performance taking into account all individual studies.

**Figure 5 diagnostics-13-00807-f005:**
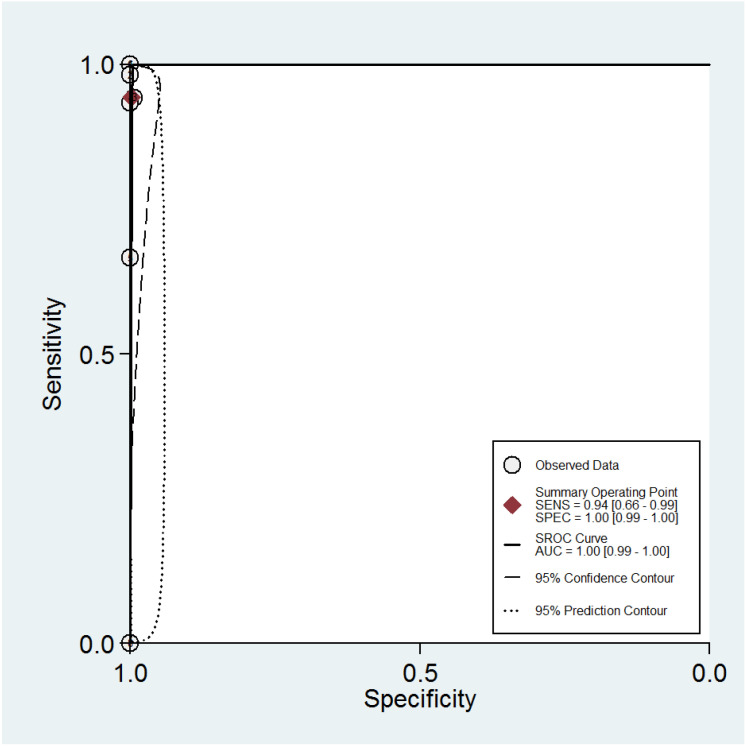
Summary ROC curve for the diagnostic performance of two-dimensional sonohysterography to detect uterine septum, showing the sensitivity and specificity for each study and pooled estimation. The dashed line around the summary point estimate (red diamond) represents the 95% confidence region. The dotted line showing the 95% prediction contour corresponds to the predicted performance taking into account all individual studies.

**Figure 6 diagnostics-13-00807-f006:**
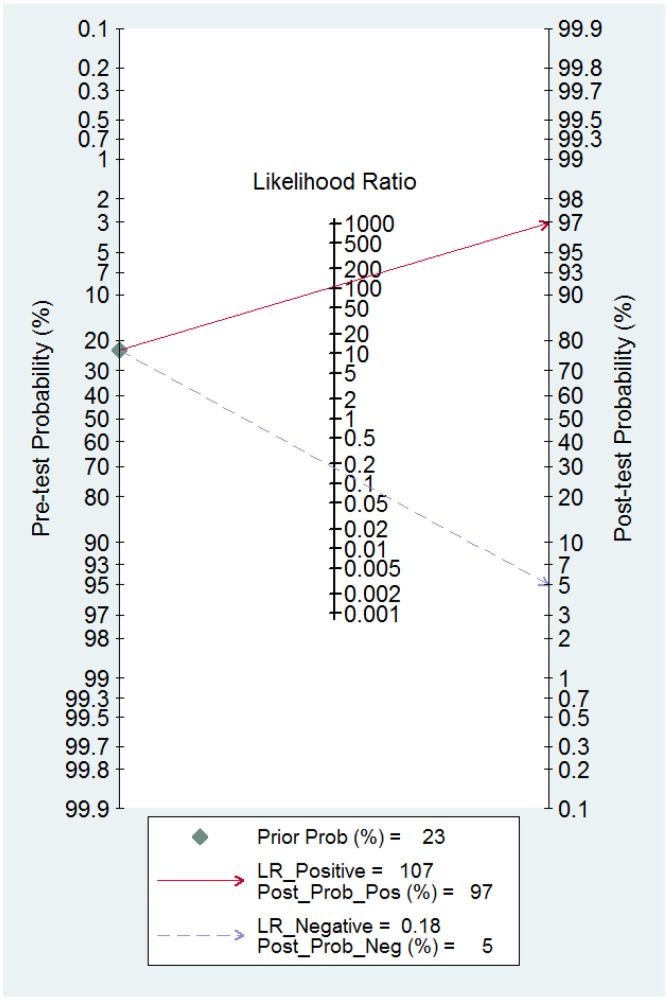
Fagan nomogram for two-dimensional transvaginal ultrasonography. It can be observed how the test changes the pre-test probability depending on a positive or negative result.

**Figure 7 diagnostics-13-00807-f007:**
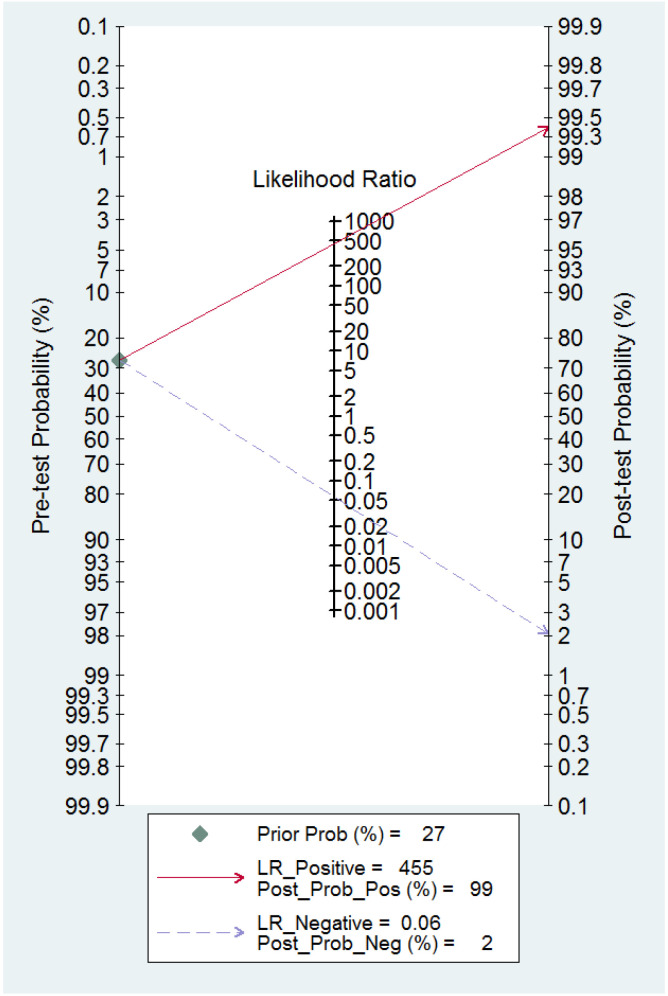
Fagan nomogram for two-dimensional transvaginal sonohysterography. It can be observed how the test changes the pre-test probability depending on a positive or negative result.

**Figure 8 diagnostics-13-00807-f008:**
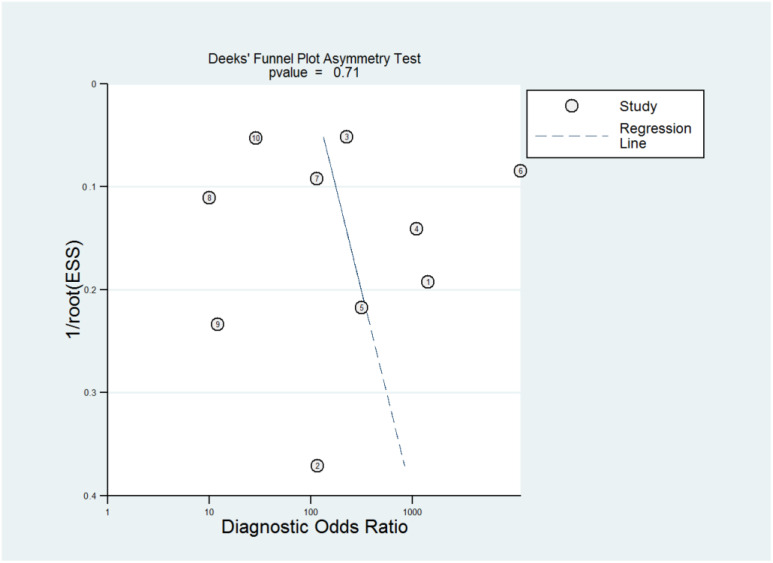
Publication bias regarding two-dimensional transvaginal ultrasonography.

**Figure 9 diagnostics-13-00807-f009:**
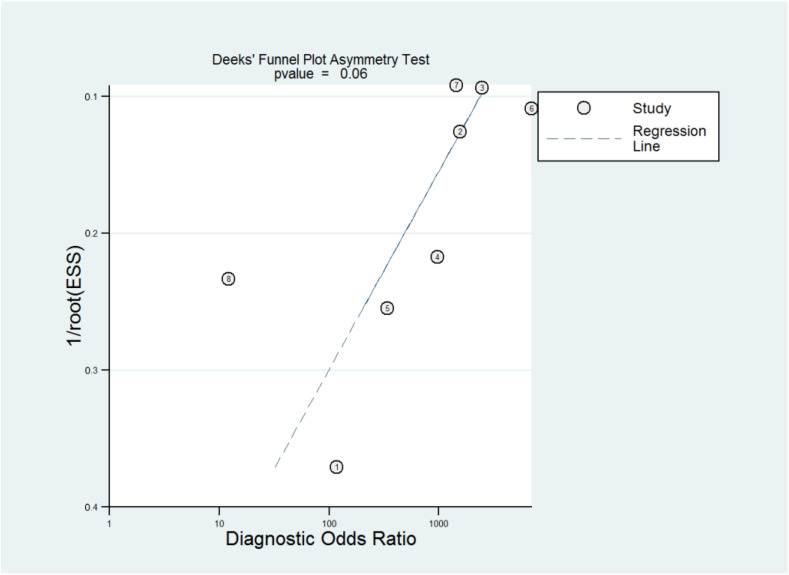
Publication bias regarding two-dimensional transvaginal sonohysterography.

**Figure 10 diagnostics-13-00807-f010:**
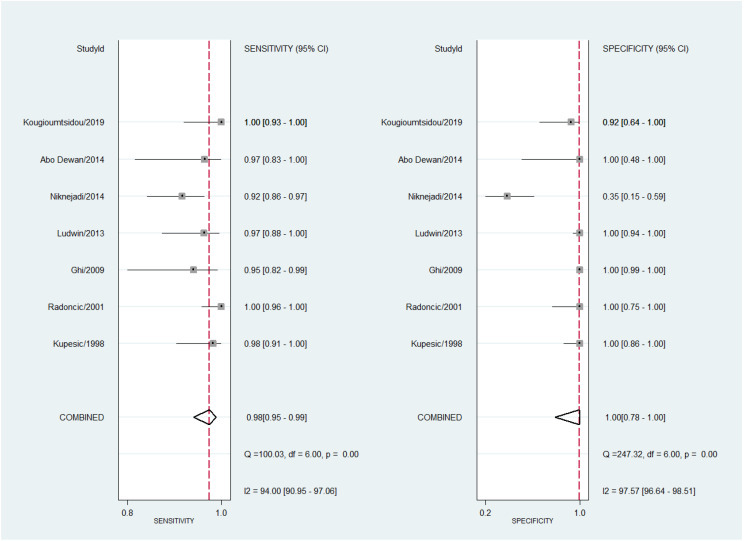
Forest plot for sensitivity and specificity for all studies concerning the diagnostic performance of three-dimensional transvaginal ultrasonography for uterine septum detection [13,24,25,30,34,35,38].

**Figure 11 diagnostics-13-00807-f011:**
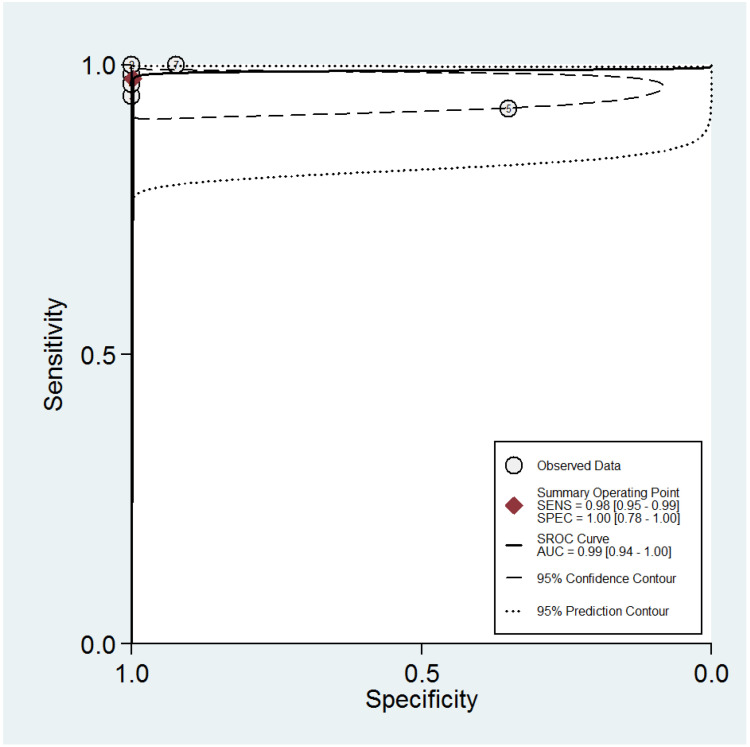
Summary ROC curve for the diagnostic performance of three-dimensional transvaginal ultrasonography to detect uterine septum, showing the sensitivity and specificity for each study and pooled estimation. The dashed line around the summary point estimate (red diamond) represents the 95% confidence region. The dotted line showing the 95% prediction contour corresponds to the predicted performance taking into account all individual studies.

**Figure 12 diagnostics-13-00807-f012:**
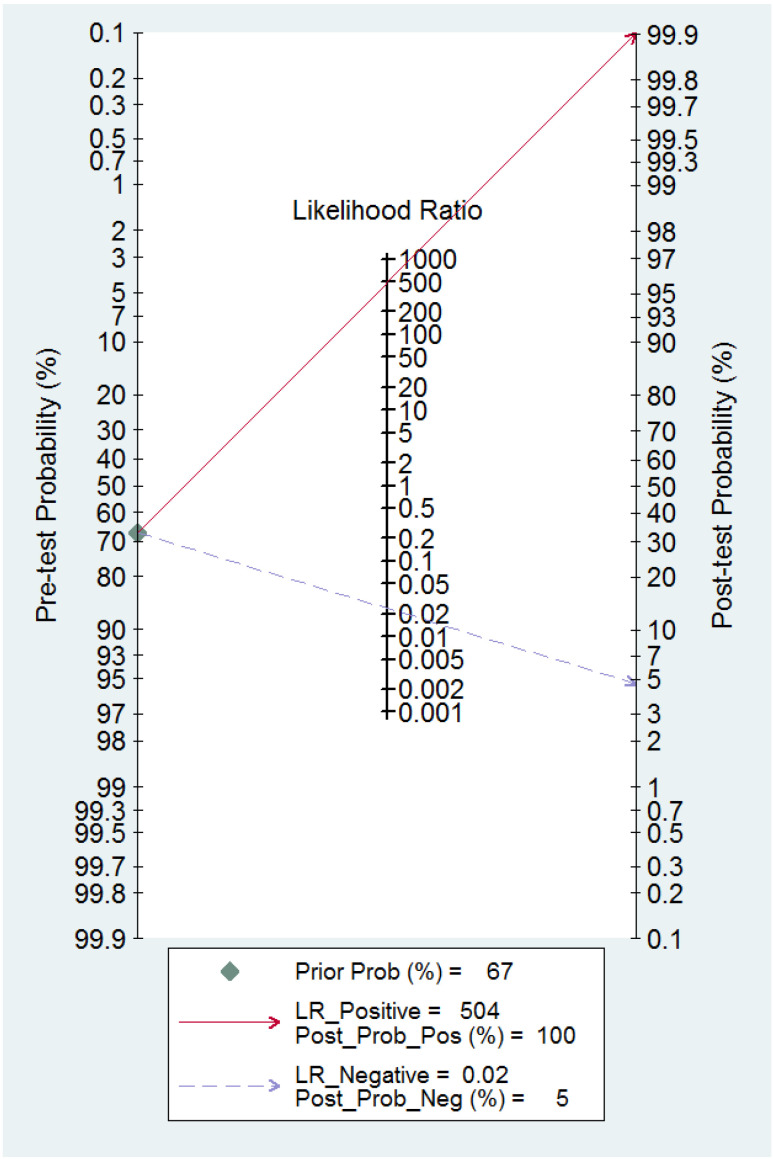
Fagan nomogram for three-dimensional transvaginal ultrasonography. It can be observed how the test changes the pre-test probability depending on a positive or negative result.

**Figure 13 diagnostics-13-00807-f013:**
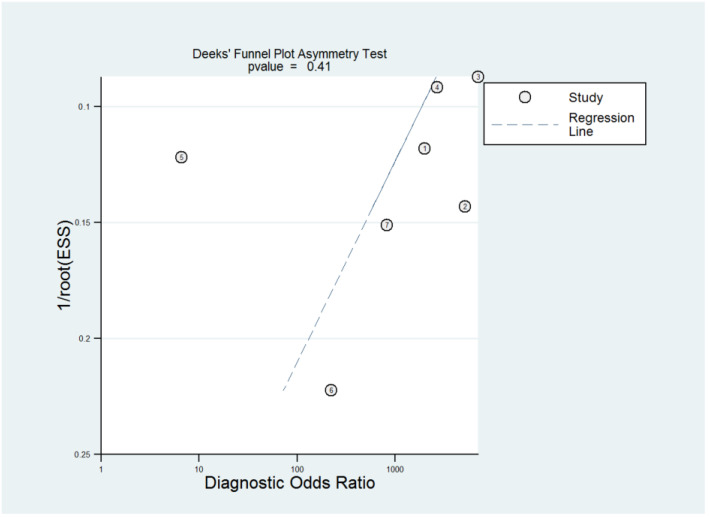
Publication bias regarding three-dimensional transvaginal ultrasonography.

**Table 1 diagnostics-13-00807-t001:** Main characteristics of the studies included in the meta-analysis.

Author	Year	Country	Study Design	Index Test	Reference Test	Uterine Septum Definition	Examiner Blinded
Narayan	1993	UK	NA	2D TVS	HSC	Authors’ own definition	NA
Romano	1994	Italy	Retrospective	2D TVS, 2D SIS	HSC	Own definition	Yes
Kupesic	1998	Croatia	NA	2D TVS, 2D SIS, 3D TVS	HSC + LPS	ASRM	NA
Radoncić	2000	Croatia	NA	3D TVS	HSC	NA	NA
Loverro	2001	Italy	NA	2D TVS	HSC	NA	NA
Alborzi	2003	Iran	Prospective	2D SIS	HSC + LPS	Own definition	Yes
Menada	2006	Italy	Retrospective	2D TVS, 2D SIS	HSC +/− LPS	Own definition	NA
De Feuce	2009	Italy	NA	2D SIS	HSC +/− LPS	NA	NA
Ghi	2009	Italy	Prospective	3D TVS	HSC +/− LPS	ASRM	Yes
Ludwin	2011	Poland	Prospective	2D SIS	HSC + LPS	Own definition	Yes
Niknejadi	2012	Iran	Cross-sectional	2D TVS	HSC	ASRM	Yes
Ludwin	2013	Poland	Prospective	2D TVS, 2D SIS, 3D TVS, 3D SIS	HSC + LPS	ASRM	Yes
Ahmadi	2013	Iran	Cross sectional	3D SIS	HSC	ASRM	NA
Niknejadi	2014	Iran	Retrospective	2D TVS, 3D TVS	HSC	ASRM	NA
Abo Dewan	2014	Egypt	Retrospective	3D TVS	HSC	ASRM	NA
Reda	2016	Egypt	Prospective	2D TVS, 2D SIS	HSC	NA	Yes
Shiva	2018	Iran	Prospective	2D TVS	HSC	Own definition	Yes
Kougioumtsidou	2019	Greece	Prospective	3D TVS	HSC + LPS	ESHRE-ESGE	Yes

NA: information not available; 2D TVS: Two-dimensional transvaginal ultrasonography; 2D SIS: 2D transvaginal sonohysterography; 3D TVS: Three-dimensional transvaginal ultrasonography; 3D SIS: 3D transvaginal sonohysterography; HSC: hysteroscopy; LPS: laparoscopy; ASRM: American Society of Reproductive Medicine; ESHRE-ESGE: European Societies of Endoscopic and Reproductive Gynecology.

**Table 2 diagnostics-13-00807-t002:** Quality assessment (risk of bias and concerns of applicability) for all studies included in the meta-analysis.

Study	Year	Risk of Bias				Concerns about Applicability		
		Patient Selection	Index Test	Reference Standard	Flow and Timing	Patient Selection	Index Test	Reference Standard
Narayan	1993	Low risk	High risk	Low risk	Low risk	Low risk	Low risk	Low risk
Romano	1994	High risk	High risk	Low risk	Low risk	High risk	Low risk	Low risk
Kupesic	1998	Low risk	Low risk	Low risk	Low risk	Low risk	Low risk	Low risk
Radoncić	2000	High risk	Unclear risk	Low risk	Low risk	Low risk	Low risk	Low risk
Loverro	2001	Low risk	Unclear risk	Low risk	Low risk	Low risk	Low risk	Low risk
Alborzi	2003	Low risk	High risk	Low risk	Low risk	Low risk	Low risk	Low risk
Menada	2006	High risk	Low risk	Low risk	Unclear risk	Low risk	Low risk	Low risk
De Feuce	2009	Low risk	Unclear risk	Low risk	Unclear risk	Low risk	Low risk	Low risk
Ghi	2009	High risk	Low risk	Low risk	Low risk	Low risk	Low risk	Low risk
Ludwin	2011	High risk	Low risk	Low risk	Unclear risk	Low risk	Low risk	Low risk
Niknejadi	2012	High risk	Low risk	Low risk	Low risk	Low risk	Low risk	Low risk
Ludwin	2013	High risk	Low risk	Low risk	Unclear risk	Low risk	Low risk	Low risk
Ahmadi	2013	High risk	Low risk	Low risk	Low risk	Low risk	Low risk	Low risk
Niknejadi	2014	High risk	Low risk	Low risk	Low risk	Low risk	Low risk	Low risk
Abo Dewan	2014	High risk	Low risk	Low risk	Unclear risk	Low risk	Low risk	Low risk
Reda	2016	High risk	Unclear risk	Low risk	Low risk	Low risk	Low risk	Low risk
Shiva	2018	High risk	Low risk	Low risk	Low risk	Low risk	Low risk	Low risk
Kougioumtsidou	2019	High risk	Low risk	Low risk	Unclear risk	Low risk	Low risk	Low risk

## Data Availability

Data are available upon reasonable request.

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
