# Peer review of "Diagnostic Performance of Two-Dimensional Ultrasound, Two-Dimensional Sonohysterography and Three-Dimensional Ultrasound in the Diagnosis of Septate Uterus—A Systematic Review and Meta-Analysis"

_diagnostics, 2023, doi:10.3390/diagnostics13040807_

Round 1

Reviewer 1 Report

The authors of the manuscript "Diagnostic performance.........."attempt to clarify the issue of diagnostic performance of three different approaches to the diagnosis of septate uterus. The introduction and the text as a whole should be edited by a native English speaker and present the results and the main conclusions in a clear and concise way.

Author Response

Thanks for your comments.

We have revised the whole manuscript for language editing

Sincerely yours

Juan Luis Alcázar 

Reviewer 2 Report

Thank you for requesting  to provide a review of this article, which has a subject of high interest. 

   The main purpose of the analysis was to perform a pooled analysis of the diagnostic performance of 2D-transvaginal ultrasonography, 2D-transvaginal sonohysterography, 3D-transvaginal ultrasound and 3D-transvaginal sonohysterography for the diagnosis of the septate uterus, even though the gold standard diagnosis for this pathology is hysteroscopy.

   The main question adressed in the research was whether 2D-sonohysterography could be a reasonable option for diagnosing the septate uterus, although 3D-transvaginal ultrasound is the best method for diagnosis.

   The study is a meta-analysis during a period of time between January 1990 and November 2022. The topic is original and relevant in the field and brings usefull knowledge regarding the subject. A comprehensive search strategy was used. The review methodology was comprehensive with screening and data extraction. When it comes to the methodology used, no specific improvements should be considered from my point of view.

   The conclusions are consistent with the evidence and the arguments presented, and they adress properly to the main question which conducted the analysis.

   The references are appropriate and well suited for this kind of study. 

    Regarding the figures and pictures used in the article, they provide suitable information about the cases and show significant statistical references. They are also well understandable and the information is easy to be followed. There are no other comments required about these items, from my point of view.

  Regarding the structure and accuracy of the phrases, the manuscript has well structured information, with supported evidence and well structured phrases.

   The manuscript is original and well defined. The results provide an advance in current knowledge. The results are being interpreted appropriately and are significant, as well as the conclusions.

  The article is written in an appropriate way. 

  The study is correctly designed and the analysis is being performed at high standards, so the data are robust enough to draw the conclusion. 

   Surely the paper will attract a wide readership. 

   The English language is appropriate and well understandable.

   There are a few things to add in the lines below, but the article should be published after the corrections are made: 

Line 60: „.” After  „miscarriage [3]”

Line 68: „.” After „explanations [6]”

Line 98: „.” After „ultrasound [14,15]”

Line 115: „.” After „guidelines [17,18]”

Line 195: „.” After „method [21]”

Line 198: „.” After „significant”

Line 236: only 1 space between „nine” and „studies”

Line 238: „.” After „[22,24-26,28,29,33-35]”

Line 498: „.” After „unclear”

Line 519: „.” After „specificity”

Line 531: „.” After „studies [40-43]”

Author Response

Dear Reviewer

Thank you very much for all your comments. We really appreciate them.

We have amend all mistakes you highlighted. Modifications marked in yellow.